# The impact of mass drug administration expansion to low onchocerciasis prevalence settings in case of connected villages

**Anneke S. de Vos**[ID]\*, **Wilma A. Stolk**[ID], **Luc E. Coffeng**[ID]⍟, **Sake J. de Vlas**[ID]⍟

Department of Public Health, Erasmus MC, University Medical Center Rotterdam, Rotterdam, The Netherlands

⍟ These authors contributed equally to this work.
\* a.s.devos@erasmusmc.nl

## Abstract

### Background

The existence of locations with low but stable onchocerciasis prevalence is not well understood. An often suggested yet poorly investigated explanation is that the infection spills over from neighbouring locations with higher infection densities.

### Methodology

We adapted the stochastic individual based model ONCHOSIM to enable the simulation of multiple villages, with separate blackfly (intermediate host) and human populations, which are connected through the regular movement of the villagers and/or the flies. With this model we explore the impact of the type, direction and degree of connectedness, and of the impact of localized or full-area mass drug administration (MDA) over a range of connected village settings.

### Principal findings

In settings with annual fly biting rates (ABR) below the threshold needed for stable local transmission, persistence of onchocerciasis prevalence can well be explained by regular human traffic and/or fly movement from locations with higher ABR. Elimination of onchocerciasis will then theoretically be reached by only implementing MDA in the higher prevalence area, although lingering infection in the low prevalence location can trigger resurgence of transmission in the total region when MDA is stopped too soon. Expanding MDA implementation to the lower ABR location can therefore shorten the duration of MDA needed. For example, when prevalence spill-over is due to human traffic, and both locations have about equal populations, then the MDA duration can be shortened by up to three years. If the lower ABR location has twice as many inhabitants, the reduction can even be up to six years, but if spill-over is due to fly movement, the expected reduction is less than a year.

**Data Availability Statement:** All relevant data are within the manuscript and its Supporting Information files.

**Funding:** The authors (ASdV, WAS, LEC, SJdV) gratefully acknowledge funding of the NTD Modelling Consortium by the Bill and Melinda Gates Foundation (OPP1184344, https://www.gatesfoundation.org/). LEC further acknowledges funding from the Dutch Research Council (NWO, grant 016.Veni.178.023, https://www.nwo.nl/en/about-nwo). The funders had no role in study design, data collection and analysis, decision to publish, or preparation of the manuscript.

**Competing interests:** The authors have declared that no competing interests exist.

## Conclusions/Significance

Although MDA implementation might not always be necessary in locations with stable low onchocerciasis prevalence, in many circumstances it is recommended to accelerate achieving elimination in the wider area.

## Author summary

When infected by onchocerciasis worm parasites, people can eventually develop blindness or severe skin morbidity. Over the past decades, in most places with high onchocerciasis prevalence, annual mass drug administration has become freely available for all inhabitants, regardless of their infection status. This policy has been highly successful in decreasing morbidity. For the next aim, to eliminate onchocerciasis, this intervention is now being expanded to lower prevalence locations. We have adapted an existing simulation model of the spread of onchocerciasis to allow us to model settings where multiple villages are connected, through movement of either humans or blackflies, the intermediate host. By this connection, worms could spill over from a high prevalence village to neighbouring villages with lower prevalence. For such situations, we have examined the impact of implementing treatment only in the high prevalence village, or also in one or two lower prevalence villages. We conclude that for elimination of onchocerciasis transmission, treatment in the lower prevalence villages may not actually be needed, but the total duration of mass drug administration in the entire area can be significantly decreased by expanding treatment to these villages.

## Introduction

The existence of locations with low but stable onchocerciasis prevalence is not well understood. Given the sexual reproduction of parasitic worms, mathematical models of their spread within a closed population (say a village) predict a breakpoint prevalence level [1–3]. Below this level, spread cannot be sustained, as female worms then often find themselves in a host without a male worm. This lack of mating opportunity decreases the prevalence and thereby the chance of mating further, eventually leading to complete population collapse.

Quantitative insight into this breakpoint level can be gained from sufficiently detailed model simulation studies. ONCHOSIM, an individual-based simulation model of onchocerciasis transmission and control, has been used for over thirty years to help inform policy [4–8]. It simulates individual human hosts as well as their residing individual worms. To enable a comparison with field observations of prevalence, it simulates skin snip surveys among the inhabitants of a village, taking into account the chance process of observing worm microfilariae (mf) in the skin. At frequently used population assumptions, ONCHOSIM predicts the breakpoint below which prevalence is unstable at around 40% mf prevalence, and by assuming intensive heterogeneity in individual exposure risk, the breakpoint can be shifted down to about 30% prevalence [9].

In reality, however, locations with low prevalence do exist [10, 11]. In the past decades, such situations, with below 40% prevalence, were not targeted by interventions against onchocerciasis [12], as programmes originally aimed for morbidity control. Severe morbidity (blindness) occurs only in high-endemic settings, when people suffer high worm load. Mass drug administration (MDA) in these high prevalence locations has been highly successful in

decreasing prevalence and morbidity [13, 14], and the focus of interventions has now shifted to elimination of transmission, which is thought to require encompassing also lower prevalence locations [15]. To understand whether and in which low-endemic settings MDA is going to be necessary for elimination, it is important to better understand how prevalence is sustained here.

One proposed explanation for the existence of stable low prevalence settings is that worm infections spill over from neighbouring high prevalence locations where onchocerciasis can sustain itself [15]. We have adapted the ONCHOSIM model to now include multiple villages, with different levels of contact between them, by regular human and/or by fly movement, allowing us to investigate the consequences of such connectivity. In particular, we aim to represent a village with high fly biting rate, say near a river where flies breed, with one or two neighbouring villages more inland, with much lower fly densities, such that in the latter onchocerciasis prevalence would not be stable if those villages were fully isolated. Due to the movement of worms (in humans) or worm larvae (in flies) between villages, however, onchocerciasis prevalence spills over from this first village and becomes established in the surrounding villages, at a lower prevalence. We therefore refer to these locations as the source and the spill-over villages, respectively.

The ultimate goal of our analysis is to explore which MDA strategy is recommended in such a setting: treating only the source village or also those villages with spill-over prevalence? We explore this question under a range of geographical settings, among others differing in annual biting rates, connection type (human or fly), and the direction and frequency of movement.

## Methods

### Basic model

ONCHOSIM is an established model of the spread and control of onchocerciasis [4–8]. It simulates the life-histories of humans as well as individual male and female worms within these humans. Patent female worms in a host produce worm microfilariae (mf) only if a patent male worm is present in the same host. Flies biting human hosts take up mf, but fly uptake capacity is limited, resulting in an uptake that saturates with the within-host mf level. The annual biting rate (ABR) parameter sets the frequency with which the average adult human in the model is bitten by flies. The aggregate of all mf taken up by the modelled fly-population is represented as a homogenous cloud of infectious material associated with a given population (say a village).

As in most analyses using ONCHOSIM, prevalence refers to the outcome of a simulated skin snip survey among all villagers aged 5+, assuming two snips performed per individual. Furthermore, the human population size is kept stable, despite the assumed high birth rate standard for African rural areas, as follows: when the total population reaches more than 110% of the target population (i.e. 400 individuals), a random 10% of the population is removed, basically reflecting migration to outside the region of interest [4].

MDA is modelled as before [7], where MDA coverage refers to the yearly fraction of villagers who receive ivermectin, the standard drug for onchocerciasis control. Ivermectin is assumed to kill all mf, and to temporarily stop all new mf production. However, female worms recover their ability to reproduce within 11 months on average, but with each dose of ivermectin their fecundity is decreased permanently by 35%. For simplicity, we assume fully random participation in our current investigation.

**Model extension.** We have adapted the ONCHOSIM version that is described in our previous paper on the influence of assortative mixing [9], which is implemented in the program *R*

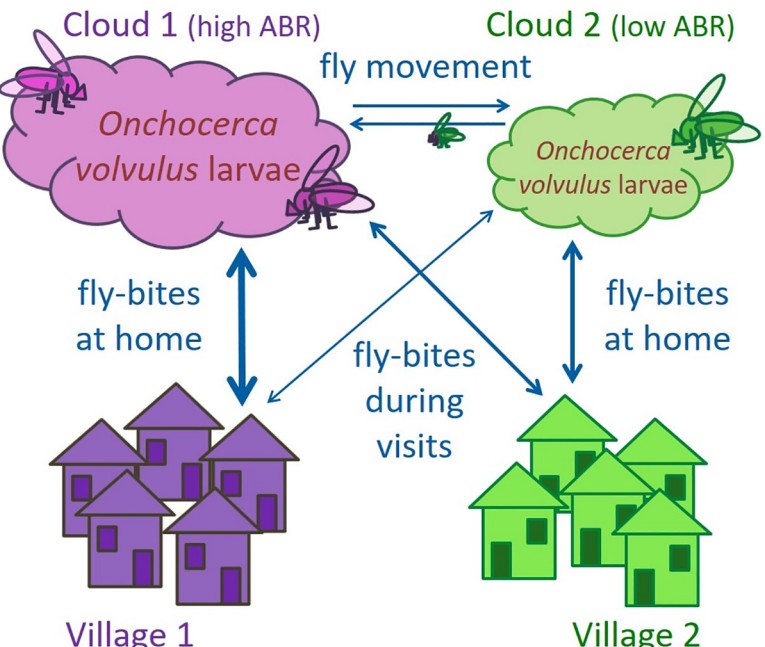

**Fig 1. Graphical representation of connectivity in ONCHOSIM.** The model keeps track of humans, individual adult *Onchocerca volvulus* worms within these humans, and the total number of worm larvae (microfilariae) that have been taken up by flies. This latter aggregate has been named the cloud. Here we model two (or three) distinct villages, each with its own cloud. Individuals spend most of their time at their home village, where through fly-bites they become infected and add to the infection-load of worm larvae in the local village cloud. The greater the annual fly biting rate (ABR) in their village, the greater the frequency of this cloud-human contact. However, individuals may also spend a proportion of their time in another village, and during this time they are infected from and add larvae to the local cloud there, with a frequency depending on the ABR in this area. The model also allows for movement of flies (with the larvae inside them) directly from cloud to cloud.

[16]. For full detail on how we have adapted the model, see Supplementary information S1 Text section : *Detailed methods with mathematical formulae*. Here we give a summary denoting our main model assumptions.

The adapted model now includes multiple (two or three) villages, which can be targeted separately with MDA. Each village has its own fly population ('cloud'). We allow for different levels of contact between the villages, either by regular human or by fly movement (see Fig 1 for a visual description). Regular human movement between villages is modelled as follows: each individual has a home village, but is also randomly assigned a fraction of time that he or she spends in each other village (drawn from the Dirichlet distribution, a multivariate generalization of the beta distribution), to be temporarily exposed to the fly cloud of that village. On average the inhabitants of a village always spend most of their time in their home village, but some individuals might even spend most time in a neighbouring village, for example where they work, rather than in their home village. Still, their infection status counts for their home village prevalence level, they can only receive treatment when MDA is implemented in their home village, and an individual's outward mobility is not associated with their probability for receiving treatment.

Fly movement is modelled by allowing a fraction of flies to move directly from one cloud to another. Note here the implicit assumption that movement between villages does not affect the probability for infectious flies to cause infection, as their rate of biting humans stays constant if they move. In our current investigation, we allow only stable transit (i.e. constant throughout a

simulation) of humans or flies between villages. Thus, we assume stable local fly population sizes.

## Model analyses

In our study we consider two or three villages, each consisting of around 400 individuals. We first study how onchocerciasis prevalence levels would depend on the extent and the type of connection between villages, considering either regular human or fly exchange, and either symmetric or unidirectional movement. We also vary the ABRs of the villages: while in source Village 1 this value is always above breakpoint level, i.e. onchocerciasis prevalence would be sustained here in case of isolation, conversely in spill-over Village 2 the ABR is below breakpoint level (and if included, in spill-over Village 3 the same or even lower), such that here onchocerciasis would disappear. In these explorations of the connectivity effects on prevalence, per setting, we perform 40 runs of 150 years each, of which 130 years are a burn-in period to achieve equilibrium conditions, and averaging over the final 20 years (monthly values) of the runs to obtain prevalence level expectations. The mid 80% range (minus the lowest and highest 10%) over the aggregated data points within the same 20-year period of all runs is given to indicate the impact of stochastic variation.

Next, we use our model to analyse MDA strategies in case of connected villages. We explore the impact of different MDA implementations in fourteen representative geographical scenarios (of which ten with two, and four with three villages), which differ regarding the village ABRs, connection type and intensity between the villages. In additional scenarios, we investigated the impact of lower or much lower MDA coverage, and higher or lower individual variation in the time spent in the other village. In each scenario, MDA is implemented in either the total area, or only in the higher prevalence location(s). In two of the geographical scenario settings with two villages, we also explore all variations in which MDA is implemented also in Village 2, but for fewer years than in Village 1, and with the same end year of MDA in both villages. We perform 100 runs per MDA setting, and the 100 initiating random seeds are kept constant for each setting, i.e. the starting conditions before MDA implementation are the same, which reduces the effect of randomised model events on the estimated impact of MDA implementation.

Preliminary inspection of runs showed that at 100 years post MDA, onchocerciasis prevalence in a modelled area has always either recovered to (near) pre-MDA levels, or it has become 0. We consider different durations of MDA, noting in how many runs elimination is achieved at 100 years post MDA. The number of years of MDA needed for 90% probability of elimination is then estimated by fitting a logistic regression model to the elimination fractions, using splines (see also Supplementary materials, S1 Text section).

Supplementary information Table A in S1 Text, *PRIME-NTD checklist guiding principles for policy relevant modelling*, provides a description of how we adhere to the five principles of the NTD Modelling Consortium [17].

## Results

### Connectivity effects on prevalence in two connected villages

We first consider two equal-sized villages to be connected by regular human movement (Fig 2). These villages differ in their local annual rate of fly biting (ABR). For example, with ABR 20,000 in Village 1 and 4,000 in Village 2, onchocerciasis prevalence would be about 88% in Village 1, and 0% in Village 2 if the two villages are isolated (top-middle panel). However, with people regularly moving between both villages, prevalence spills over from Village 1 to Village 2; if the average individual spends 10% of the time in the other village, onchocerciasis

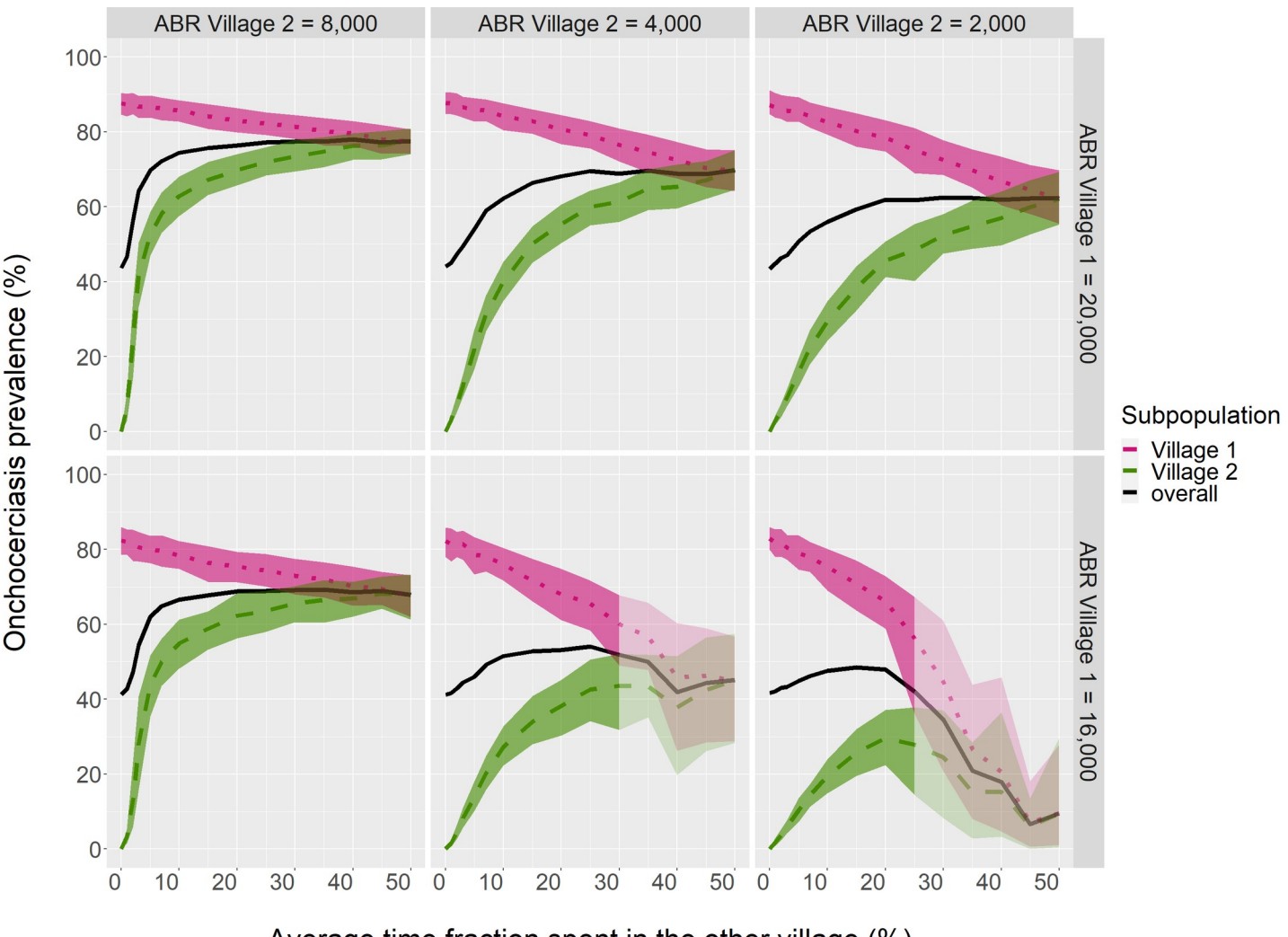

**Fig 2. Spill-over equilibrium prevalence as influenced by the local annual biting rates (ABRs) of two connected villages.** Movement is human and symmetric between the villages. Bands represent the 80% inter-decile range over the aggregate of monthly data-points for all runs. Transparency shows where 1–20 of the 40 runs ended with onchocerciasis eliminated; mean and variance are then calculated over remaining runs only, and lines are cut off where >20 runs had onchocerciasis disappear. Prevalence is among those aged 5+ years.

prevalence in Village 2 increases to 40%, while due to this connectivity prevalence in Village 1 is lowered by 4%-points to 84%.

With the ABR in Village 2 doubled, at 8,000, prevalence here increases much more rapidly with increasing level of connection to Village 1; even when keeping the overall biting rate, summed over both villages, equal (i.e. Village 1 ABR lowered from 20,000 to 16,000, Fig 2 lower-right panel), 44% onchocerciasis prevalence in Village 2 is now reached with inhabitants on average spending just 5% of their time in the other village, with prevalence in Village 1 lowered by 2%-points in this case. At 10% human exchange, Village 2 prevalence in this setting reaches 55%, while Village 1 prevalence is lowered by 4%-points. Since the ABR in Village 2 is now near the threshold where onchocerciasis could be sustained if Village 2 were isolated (in this model at ABR ≈ 10,000), there can be significant spread through the flies of Village 2 once onchocerciasis has become established here through spill-over.

Conversely, with lower ABR in Village 1, spread of onchocerciasis here may be destabilised by high connectivity to Village 2; with ABR 16,000 in Village 1 and 4,000 in Village 2 (lower-middle panel), and individuals spending on average 25% or more of their time in the other village, the population of worms becomes spread out too thinly over all villagers, leading to a lack of mating opportunity, and thereby eventual elimination of onchocerciasis in the whole area.

It is especially the people from the low biting rate area visiting the higher biting rate area which allows for the spill-over of prevalence (see Fig 3, top row). When Village 1 inhabitants visit Village 2, they are exposed to a much lower biting rate, so that they are not as likely to transmit their infection to others, while conversely the visitors from Village 2 to Village 1 are relatively highly exposed during their stay. When the villages are connected by movement of flies, this has a rather similar effect on the prevalence levels in the connected villages as by

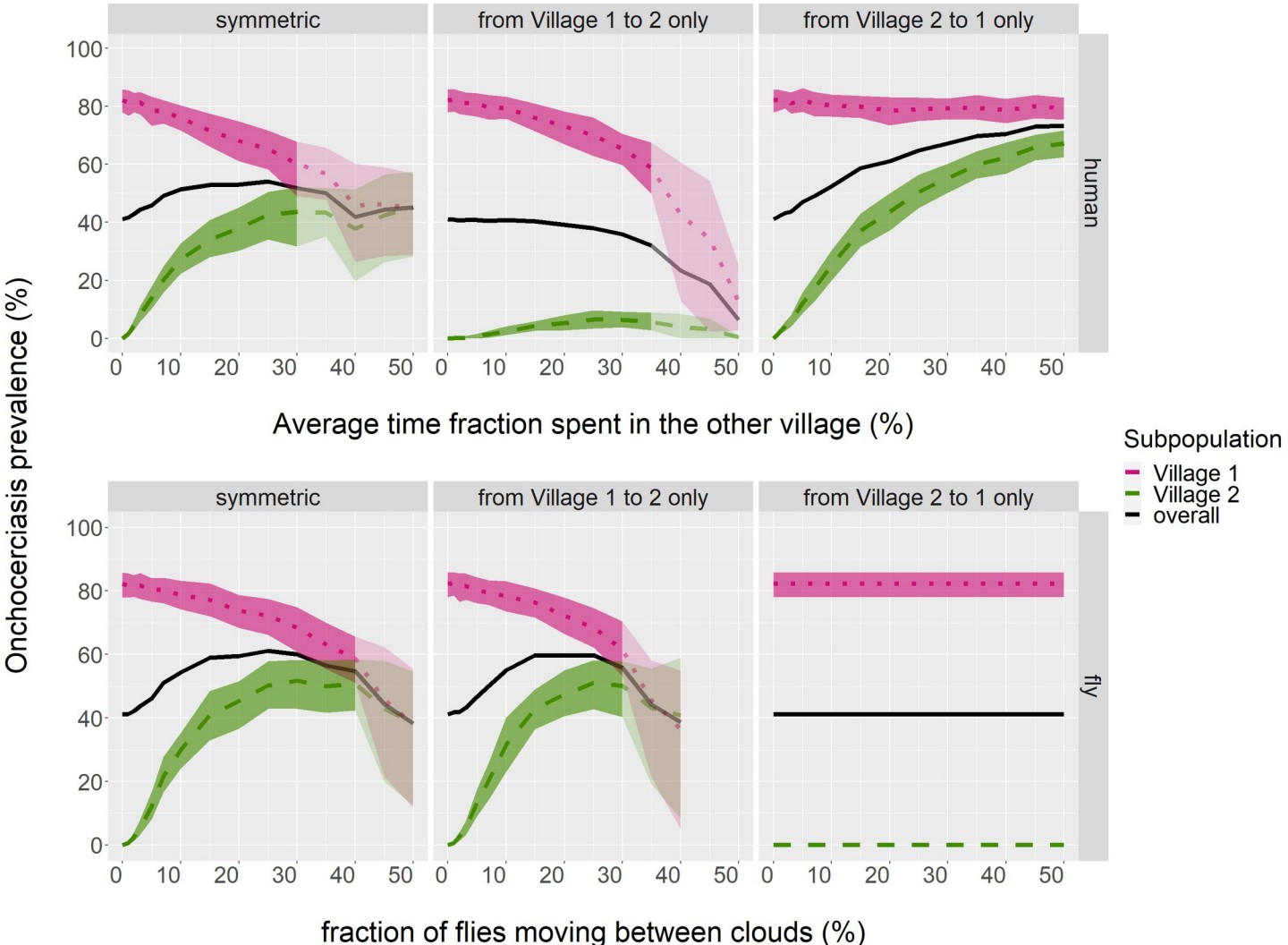

**Fig 3. Spill-over equilibrium prevalence as influenced by the type of connection between two villages.** ABR Village 1 = 16,000; ABR Village 2 = 4,000. Movement is either symmetric, i.e., the same number of inhabitants or flies travel to Village 1 from Village 2 as vice versa (left panels), or travel occurs only by those from one of the two villages (middle and right panels). The average time fraction in the other village, or the fraction of flies moving between clouds parameter, is set only for those that do travel, i.e. symmetric movement here represents the addition of the other two types of travel. Bands represent the 80% inter-decile range over the aggregate of monthly data-points for all runs. Transparency shows where 1–20 of the 40 runs ended with onchocerciasis eliminated; mean and variance were then calculated over remaining runs only, and lines are cut off where >20 runs had onchocerciasis disappear. Prevalence is among those aged 5+ years.

human movement (Fig 3, bottom row). However, movement of flies may be more strictly one way, for example due to the wind blowing the flies exclusively from one to the other village. Flies can then cause a significant spill-over prevalence if they move from the source to the spill-over village, yet in the reverse direction their movement would have no effect at all, since flies are not expected to return home, carrying their infection back, as villagers do.

When both humans and flies move between villages, the two types of connectivity simply add up their effects (see Fig A in S1 Text). Other geographical dimensions may obviously differ for a particular setting in the actual world, and we have therefore also considered alternative ABR combinations, villages of different relative sizes, and a greater number of villages (Figs B, C and D in S1 Text). These factors do not change the general shape of spill-over prevalence by connectivity between villages, but have quantitative effects. We also considered more or less heterogeneity between individuals in their tendency to go outside of their home village (Fig E in S1 Text), and find this factor not to have much effect: it is the sum amount of connection between the village clouds of intermediate hosts, rather than individual movement behaviour that causes spill-over of prevalence.

## MDA strategies in case of two connected villages

Fig 4 gives an illustration of MDA in case of two connected villages, source Village 1 (ABR = 16,000) and spill-over Village 2 (ABR = 8,000). Since presence of onchocerciasis is self-sustaining only in Village 1, sufficiently long MDA in this source location alone will eventually lead to elimination of onchocerciasis in Village 2 as well. However, after some years of MDA, residual prevalence levels in the untreated spill-over village will become higher than in the treated source village. If MDA is stopped too soon, export of infection from Village 2 may ignite resurgence of onchocerciasis in Village 1 (left panels). Yet if also the spill-over village receives treatment, prevalence here is rapidly lowered as well, preventing reverse spill-over from Village 2 to Village 1, making resurgence much less likely (right panels).

To more broadly explore the effect of different MDA strategies in case of connected villages, we have chosen a varied set of fourteen geographical settings (Table 1). For example, with ABR 16,000 and 8,000 in Village 1 and 2 respectively, and 3% exchange of the villagers, treating only Village 1 inhabitants it takes 12.5 years of MDA to reach 90% certainty of onchocerciasis elimination, but treating all villagers this takes only 11.1 years, a gain of 1.4 years (see Fig 5). If villagers spend on average 5% of their time at the other village, the gain in decreased MDA duration from also treating Village 2 inhabitants becomes 2.8 years.

With spill-over due to human movement between two villages of near equal size, the decrease in the needed MDA duration, by expanding treatment to the lower prevalence village, is largely predictable given the pre-MDA prevalence in this spill-over village (Fig 6). About one year less MDA is needed when prevalence in Village 2 is around 25% pre-MDA, and this becomes around three years when the pre-MDA prevalence in the spill-over area is around 40%. At even higher starting prevalence in the spill-over area, the potential gain by treatment in Village 2 rises quickly, to almost a decade shortened duration at 50% pre-MDA prevalence, but as stated before, >40% prevalence areas would already be treated under prior guidelines.

As shown above (Fig 3), the direction of human movement impacts on the expected level of spill-over prevalence. Yet for a given prevalence in Village 2, the absolute decrease effected in needed MDA duration by also including village 2 in the treatment programme, is independent of directionality, since commuters either way have the potential to cause resurgence in Village 1 (Fig 6, compare scenarios 3 and 8). When prevalence spills over by fly rather than by human movement, however, expanding treatment to the spill-over village has much less impact on the required MDA duration (Fig 6, compare scenario 4 with 9, and 6 with 10); flies moving from

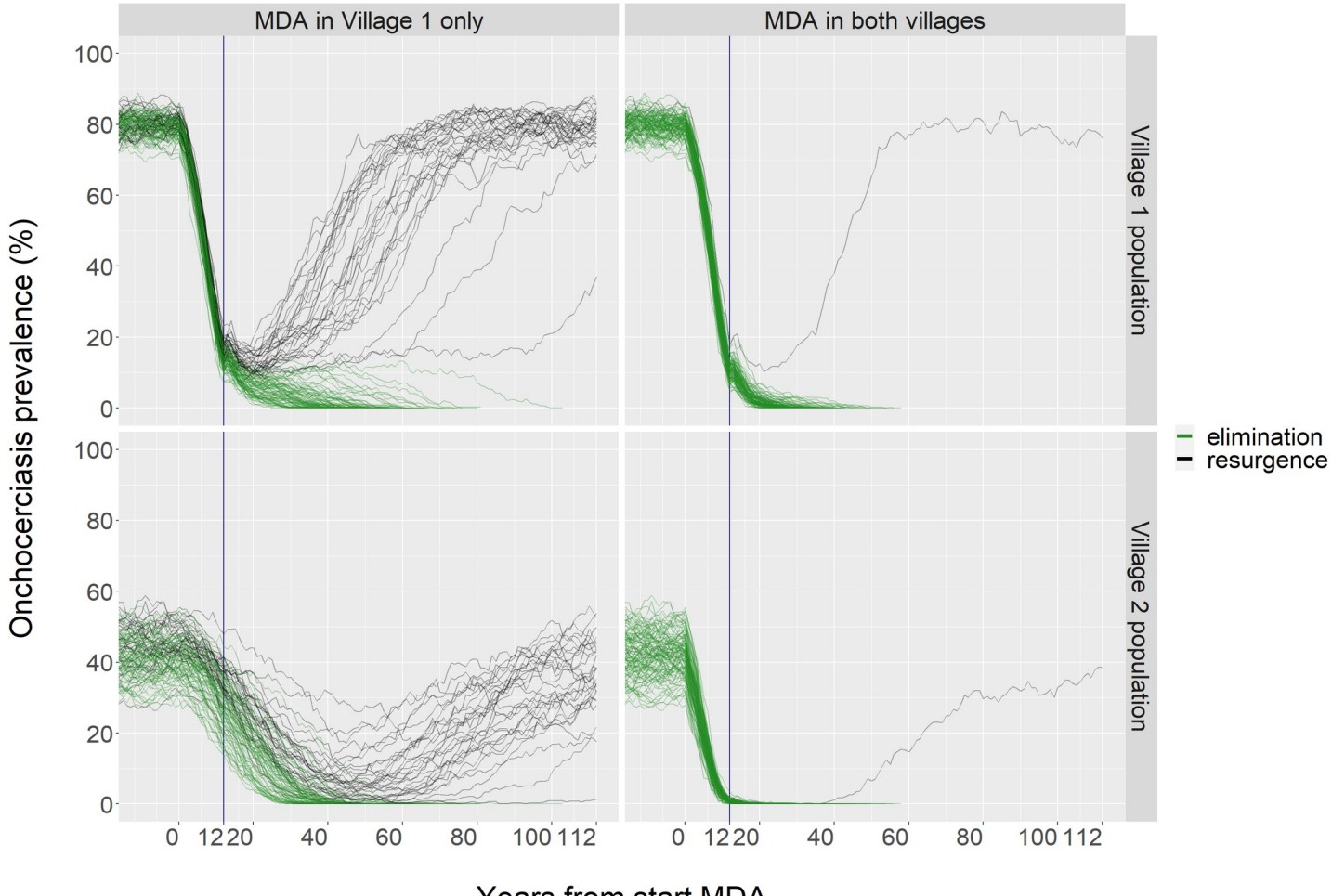

**Fig 4. Example of the impact of MDA on prevalence of onchocerciasis in two connected villages.** Each line represents one model run. ABR Village 1 = 16,000, ABR Village 2 = 8,000. On average, individuals spend 3% of their time in the other village, assuming symmetrical mixing. MDA duration is 12 years, and is either implemented in both villages or only in Village 1. Each run is continued for 100 years post MDA, at which timepoint it is determined whether either onchocerciasis prevalence is eliminated (0.0%) or whether resurgence of prevalence has occurred. With MDA in both villages, resurgence occurs in just 1 out of 100 runs, while with MDA in Village 1 only, resurgence occurs in 30 of the 100 runs. With MDA implemented in just the source location (Village 1), resurgence is to a large extent determined by the onchocerciasis prevalence in the spill-over prevalence location (Village 2) at the end of MDA; for those 30 runs showing resurgence, the mean prevalence in Village 2 at the end of MDA (vertical lines at 12 years) is 35% (range 21%-48%), while it is on average 28% (14%-44%) among the 70 runs showing eventual elimination. Prevalence is among those aged 5+ years.

spill-over Village 2 to source Village 1 are less likely to cause resurgence than humans moving in this direction, as their biting rate does not change by moving. Note also that under unidirectional fly movement, treatment in Village 2 can never affect elimination of onchocerciasis: either flies move only towards Village 2, so that remaining prevalence here cannot trigger resurgence in Village 1, or they move in the reverse direction, making spill-over impossible, as explained above.

The risk for resurgence in Village 1 to be triggered from Village 2 is only very modestly impacted by variation among villagers in their tendency to go outside of their home village (Fig J in S1 Text), again indicating that it is the total amount of connection by mobility rather than individual movement patterns that matter. In case fewer individuals are reached by MDA per village location, implementing treatment in Village 2 leads to a larger absolute decrease in the needed MDA duration to fully halt the spread of onchocerciasis, yet in relative terms the

**Table 1. Impact of MDA implementation strategies, for different geographical settings.**

| Scenario # | ABR Village 1 | ABR Village 2 | ABR Village 3 | Mean time in the other village (s) | Scenario specifics: | Pre-MDA oncho. prevalence (5 + pop.) Village 1 | Pre-MDA oncho. prevalence (5 + pop.) Village 2 | Pre-MDA oncho. prevalence (5 + pop.) Village 3 | Years of MDA for 90% elimination probability, treating Village 1 | Years of MDA for 90% elimination probability, treating Village 1 & 2 | Years of MDA for 90% elimination probability, treating Village 1, 2 & 3 |
|---|---|---|---|---|---|---|---|---|---|---|---|
| 1 | 16,000 | 4,000 | - | 7% | - | 78% | 20% | - | 10.8 | 10.4 | - |
| 2 | 16,000 | 8,000 | - | 3% | - | 81% | 28% | - | 12.5 | 11.1 | - |
| 3 | 16,000 | 4,000 | - | 15% | - | 73% | 35% | - | 10.0 | 8.1 | - |
| 4 | 20,000 | 8,000 | - | 3% | - | 87% | 41% | - | 17.9 | 14.8 | - |
| 5 | 16,000 | 8,000 | - | 5% | - | 80% | 42% | - | 13.5 | 10.7 | - |
| 6 | 20,000 | 8,000 | - | 5% | - | 87% | 52% | - | 22.6 | 14.4 | - |
| 7 | 16,000 | 4,000 | - | 15% | from Village 1 to 2 only | 76% | 4% | - | 9.5 | 9.7 | - |
| 8 | 16,000 | 4,000 | - | 15% | from Village 2 to 1 only | 80% | 36% | - | 12.4 | 10.4 | - |
| 9 | 16,000 | 4,000 | - | 15% | symmetric fly movement | 77% | 41% | - | 10.0 | 9.7 | - |
| 10 | 20,000 | 8,000 | - | 5% | symmetric fly movement | 87% | 51% | - | 17,1 | 15.1 | - |
| 11 | 16,000 | 8,000 | 4,000 | 10% | Village 2 between 1 & 3* | 77% | 34% | 11% | 11.6 | 9.7 | 9.4 |
| 12 | 16,000 | 8,000 | 4,000 | 20% | Village 2 between 1 & 3* | 70% | 41% | 21% | 10.8 | 7.3 | 6.7 |
| 13 | 16,000 | 8,000 | 8,000 | 5% | Village 1 between 2 & 3** | 80% | 36% | 36% | 15.3 | 12.5 | 10.0 |
| 14 | 20,000 | 8,000 | 4,000 | 20% | Village 2 between 1 & 3* | 81% | 55% | 33% | 19.5 | 11.8 | 10.8 |
| 15 | 16,000 | 8,000 | - | 5% | MDA coverage 65% | 80% | 42% | - | 18.7 | 14.3 | - |
| 16 | 16,000 | 8,000 | - | 5% | MDA coverage 50% | 80% | 42% | - | 24.7 | 20.7 | - |
| 17 | 16,000 | 4,000 | - | 15% | high variation movement | 73% | 33% | - | 10.1 | 8.0 | - |
| 18 | 16,000 | 4,000 | - | 15% | low variation movement | 72% | 37% | - | 9.7 | 7.8 | - |

If not otherwise stated, the villages are connected by symmetric human movement, MDA coverage is 80%, and the variance among individuals in the time they spent in the other village(s) is at baseline level (see Fig D in S1 Text). Each village has around 400 inhabitants. See for model data and fits Fig 5 (scenarios 2 and 5), and Figs F-I in S1 Text (all scenarios).

* Village 1 & 3 inhabitants spend on average one third of their time away from their home village at each other's villages, they are double as likely to visit the nearer Village 2. Village 2 inhabitants are as likely to visit Village 1 as they are to visit Village 3.

**Village 2 & 3 inhabitants spend all their time away from home at Village 1, while those from Village 1 are as likely to visit Village 2 as they are to visit Village 3.

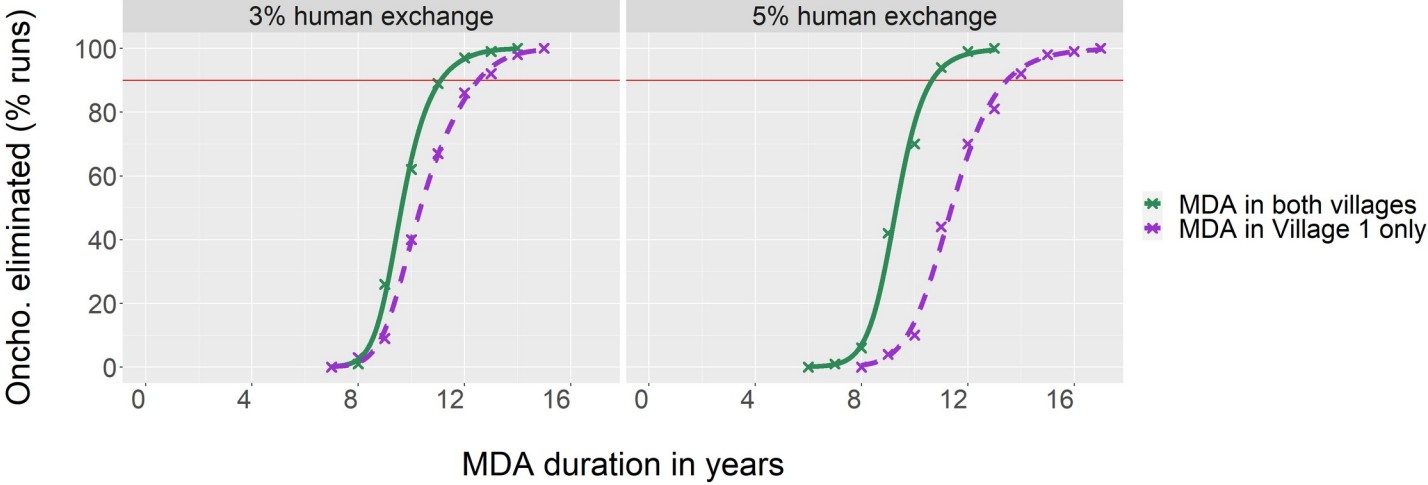

**Fig 5. Impact of MDA strategy on the probability of onchocerciasis elimination for situations with villagers on average spending respectively 3% or 5% of their time in the non-home village.** ABR Village 1 = 16,000, ABR Village 2 = 8,000. Crosses represent the proportion of runs with onchocerciasis eliminated at 100 years post MDA; lines are logistic fits to these points.

impact stays comparable, since with lower coverage more consecutive years of MDA are needed in any case (Table 1). In a third sensitivity analysis, we focus on obtaining an even higher, 99% certainty of onchocerciasis elimination. Obviously, the required MDA duration would become longer under any strategy, and we see again more impact in absolute terms from extending treatment; at 40% pre-MDA Village 2 prevalence and equal population sizes, implementation in Village 2 would then decrease the required MDA duration by four years (Fig K in S1 Text).

### MDA effects in case of three connected villages

We assume spill-over Village 3 to be as far (scenario 12) or even further (scenarios 10, 11 and 14) from the fly-breeding site(s) in or near the Village 1 location, compared to spill-over Village 2 (Table 1). When implementing MDA, there are three options; setting the prevalence threshold for starting treatment high, we treat only Village 1, setting the boundary lower, Village 1 & 2 would receive treatment, and without any boundary, the full area would receive treatment. Again, the exact geographical set-up seems to be not very important. In all four endemic settings, on a per village basis, adding treatment of Village 2 or Village 3 reduces the required number of years of MDA similarly to adding Village 2 treatment in the two-village setting, with the diminishment in the required years of MDA depending on the village's pre-MDA prevalence (Fig L in S1 Text). However, when considering Villages 2 and 3 together as one spill-over location, and using the overall prevalence in both villages as a predictor for the use of implementing treatment here, we find a doubled impact compared to when we considered a single spill-over village (Fig 6, compare diamonds and circles). This can be explained by the fact that a relatively more populous untreated spill-over area increases the risk for resurgence in the source village proportionally.

### MDA effects in case of unequal durations of MDA

Since the modelled villages have equal population sizes, expanding treatment throughout to Village 2 (and 3) would only be drug-dose saving if the total number of required MDA rounds would then be less than half (or a third) of that needed for treating only Village 1. This is not

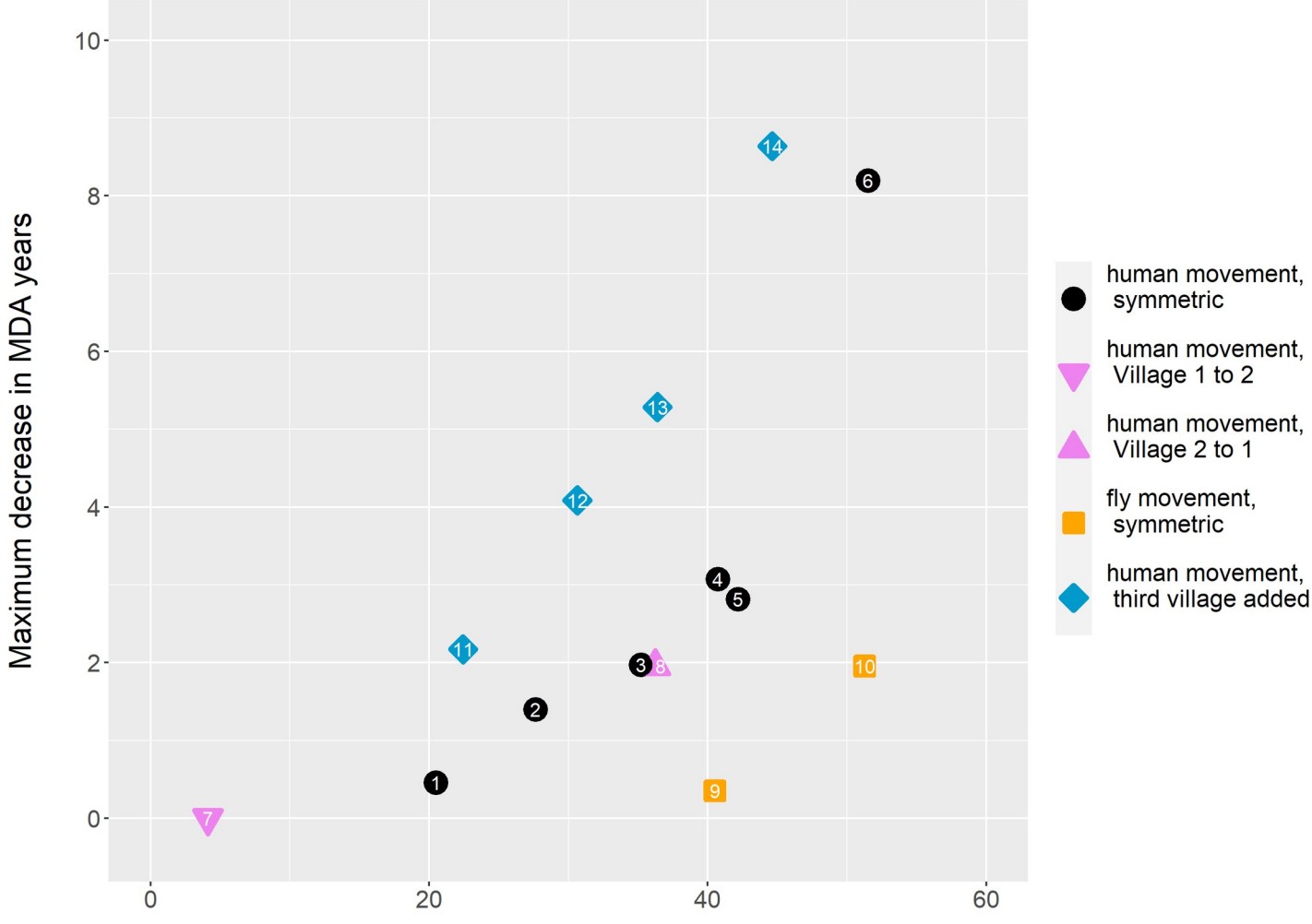

**Fig 6. Decrease in the number of years of MDA required for 90% onchocerciasis elimination probability in an area with two or three connected villages, by treatment implemented also in the spill-over prevalence village(s), rather than just in the higher prevalence village.** This is the maximum obtainable decrease, that is gained by treatment in the spill-over location(s) throughout. Numbers indicate the scenario settings, as described in Table 1. Prevalence is among those aged 5+ years. For the three village settings (blue markers), the average prevalence of Villages 2 and 3 is indicated on the x-axis, and on the y-axis the decrease in required duration by additional treatment in both of these villages (alternative comparisons with three villages are shown in Fig L in S1 Text). (For scenario 7, due to stochastic model results, a slightly negative estimate resulted (-0.2), here set to zero, since logically additional MDA should not lead to slower elimination).

the case in any of the considered settings (Table 1). However, since the starting prevalence is lower here, treatment in the spill-over village(s) may not need to be implemented for the same duration as in the source village. For this reason, but also since in most locations with meso or hyper endemicity (>40% prevalence) MDA is currently already many years ongoing, for two of the geographical settings (scenarios 2 and 5) we consider also all possible MDA duration combinations where MDA starts later in Village 2, keeping the end year of MDA concurrent in the two villages (see Fig M in S1 Text for full results).

We find that late implementation of MDA in Village 2 can still have a significant impact on the total required MDA duration for onchocerciasis elimination (Fig 7). For example, in geographical scenario 5, with pre-MDA prevalence at 80% and 42% in Village 1 and 2 respectively, by implementing MDA only in Village 1 it takes 13.5 years of MDA to achieve onchocerciasis

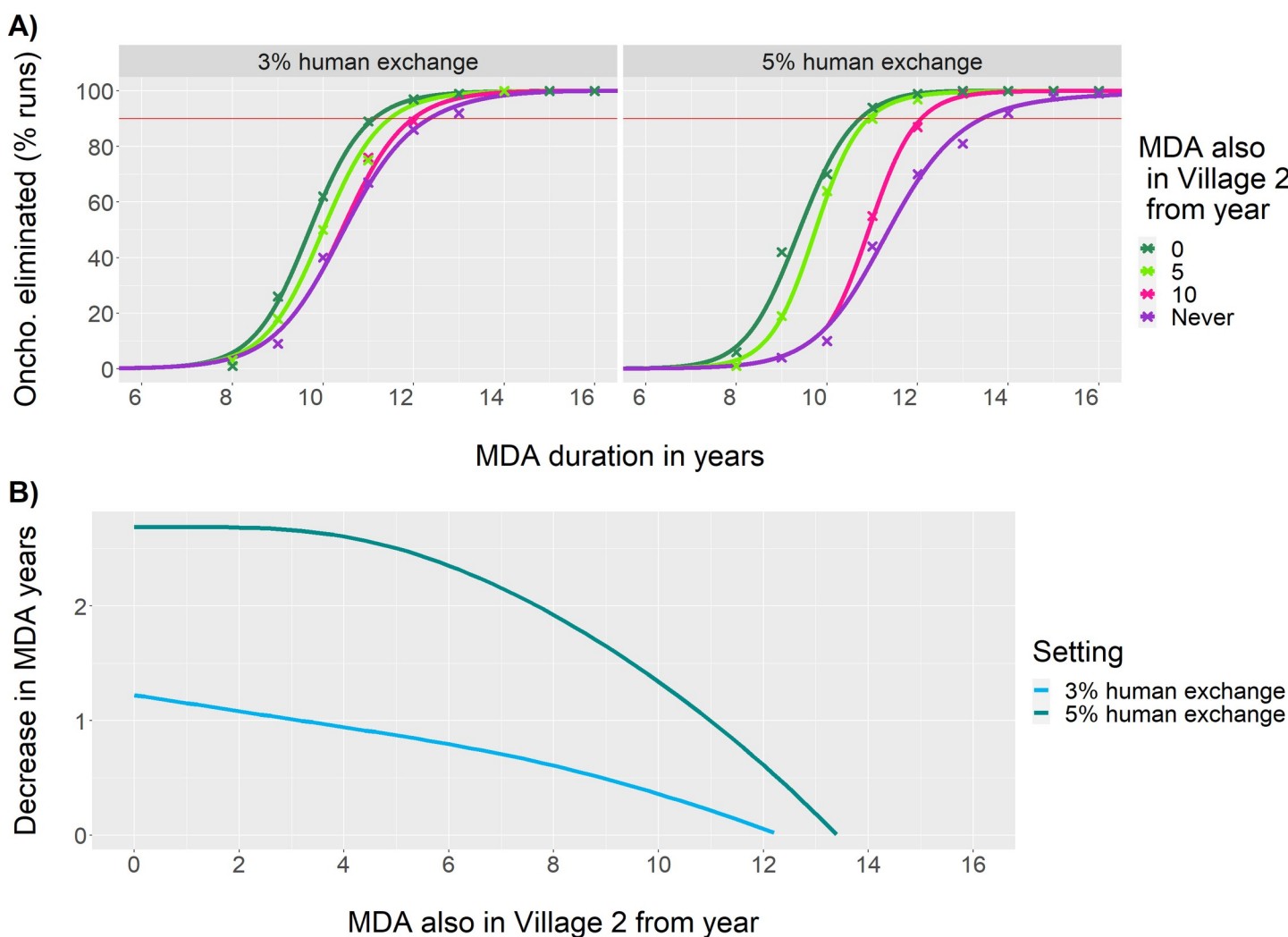

**Fig 7. Impact of MDA duration on onchocerciasis elimination when treating inhabitants of spill-over Village 2 fewer years than those of source Village 1, assuming that villagers spend 3% or 5% of their time in the non-home village.** ABR Village 1 = 16,000, ABR Village 2 = 8,000. MDA is started at the same time or later in Village 2, but always the end year of MDA in the two villages is concurrent. A) Crosses represent the proportion of runs with onchocerciasis eliminated at 100 years post MDA; lines are logistic fits to these points, see Methods. B) The corresponding fitted estimate of decrease in MDA duration by also treating Village 2 inhabitants.

elimination with 90% certainty, but if after ten years in Village 1 MDA is started also in Village 2, a total duration of only 12.0 years of MDA is needed. If we start treatment in Village 2 five years after treatment began in Village 1, we gain almost the same as if we had implemented MDA throughout in both villages; 10.9 years of MDA are then needed for 90% certainty of onchocerciasis elimination. However, for reaching eventual elimination of onchocerciasis, each year of treatment in Village 2 is only as or less effective than longer treatment in Village 1. Only when the starting prevalence in Village 2 is very high, at 52% (so that MDA would normally already be commenced here), treatment in Village 2 could be drug-dose saving (Fig N in S1 Text).

## Discussion

Our modelling study clearly illustrates that the existence of stable low onchocerciasis prevalence settings can well be explained by spill-over from neighbouring connected locations with

higher prevalence levels. Depending on the fly biting rates in source and spill-over villages, 5% to 20% exchange of either flies or human inhabitants can be sufficient to cause up to 40% prevalence in the spill-over village, where in isolation this location would not be able to maintain onchocerciasis. In case of connection by fly exchange, it is only the flies moving from the high to the low ABR location that cause the spill-over. For humans, conversely, assuming inhabitants return home after visiting the other village for work or social reasons, either village's inhabitants travelling can cause spill-over, but the effect for travelling is largest when people from the low ABR location visit the high ABR location. Note that in practice the higher biting rate location, near a blackfly breeding site (i.e. along a river), may often draw people from elsewhere, by offering opportunities for e.g. fishing, making it the setting most expedient for spill-over of infection.

Theoretically, treatment in dependent spill-over villages is not necessary for elimination, but treatment in these villages can shorten the necessary duration of MDA to reach elimination in the overall area. In practice, MDA will commonly already have started in higher prevalence areas (>40% mf prevalence), but still may have to be extended into surrounding low-endemic areas. Since lower prevalence areas require shorter MDA duration to obtain very low onchocerciasis levels [6, 7], it is not a great impediment that MDA is actually started later here than in the source area villages. For the two example settings in our study with mixed MDA durations (Fig 7B) we showed that almost the same shortening of total required MDA duration can be obtained when treatment in the low ABR location is started a few years after MDA is first implemented in the higher prevalence location.

Expanding MDA to the lower prevalence locations has much greater impact on the speed of onchocerciasis elimination when the locations are connected by human rather than by fly movement. Impact is also larger at higher pre-MDA prevalence in the spill-over location, and if the spill-over location has a relatively large population compared to the source location. When connected by human movement, and at the maximum mf prevalence for which MDA would not have yet or only recently be implemented, i.e. at 40%, and with equal population sizes, it would shorten the required MDA duration to reach 90% certainty of onchocerciasis elimination by three years if inhabitants of the spill-over location are also treated. If the spill-over location has twice the number of inhabitants relative to the source, this potential gain from expanding treatment rises to six years. An even larger relative spill-over area population size would increase the impact of treatment here proportionally, but such a setting could only occur with an extremely high biting rate in the source area; the outflow of worms from the source to the spill-over area would otherwise have already destabilised spread of onchocerciasis at the source, by lowering prevalence here to below the breakpoint level.

In all situations considered, except where pre-MDA prevalence in the spill-over location was >40%, expanding treatment was not a drug-dose saving strategy. Just treating the source location some years longer would require fewest drug doses. This option would also save the effort of setting up local treatment implementation. However, expanding MDA will of course more quickly eradicate the onchocerciasis health burden, as well as shorten the duration and costs of distributing drugs. A cost effectiveness assessment of the different intervention options could shed more light on this, but was beyond the scope of our study.

In this exploratory study, fly movement was modelled in a relatively simple manner, migrating a fraction of infected flies directly from one location to another each month. We did not consider the possibility of excess mortality among migrating flies, nor possible associations between migration, fly age and infection status [17]. By ignoring these finer points, we may have somewhat overestimated the potential for fly movement to cause spill-over prevalence. Measuring how far flies tend to travel is difficult and labour-intensive [17], making it hard to identify realistic ranges for the fraction of flies migrating. Since we found that extending MDA

to spill-over locations had only limited impact on the required MDA duration for the total area when the spill-over was due to fly movement, we chose to explore mostly MDA scenarios with villages connected by human mobility.

Human mobility was simulated assuming that individuals spend a limited portion of their time outside of their home village each month. Although human movement would be easier to study than fly movement, it is also generally undocumented, making it hard to quantify more elaborate mobility models. Our model could be extended to examine other forms of connection by human mobility, such as by seasonal or long-term human migration. We found that the total amount of connection by mobility between locations, rather than the individual movement patterns of inhabitants, determines spill-over and probability for resurgence. Also given the 10-year average lifespan of worms, we therefore do not expect that seasonal worker migration instead of the constant commuting rate considered here would lead to substantially different results. However, in our analyses we assumed that the individual human movement rate was not associated with MDA participation, while those spending much time away from their home village could be less likely to participate in local MDA. This would represent a form of systematic individual non-participation, as has been considered in other ONCHOSIM studies [7]. Specifically targeting migrant workers could then be considered as a treatment strategy to prevent resurgence.

Our model analyses were inspired by the extensive discussion about the World Health Organization policy guidelines regarding thresholds below which implementation of MDA is unnecessary, or can be discontinued [12, 18, 19]. Although potentially useful in shortening overall treatment duration, our analyses also indicate that it may be highly wasteful of drug resources to implement treatment in the very lowest prevalence locations. In the spill-over villages, mf prevalence could still be much higher than the proposed treatment thresholds of 1–5% [18], even though the probability for elimination reached in the area was already >90% (see Fig 4). It is important to stress however that our analyses only account for situations where stable low prevalences (<40%) are due to spill-over from higher prevalence locations.

In a previous paper, we showed how within an isolated population, extreme variation of risk, in particular assortative mixing of high-risk individuals, impacts on prevalence and prevalence stability, potentially explaining part of stable low prevalence (10–40%) situations [9]. As mechanism for this assortative mixing, we assumed that individuals who frequently visit locations with high fly density (i.e. the fly-breeding sites along rivers [10]), are not only bitten most often, but are also bitten by the same flies that bite other highly exposed individuals. The resulting concentration of worms in this subgroup of individuals conserves the opportunity for worms to mate, even in low prevalence settings. The partial mixing of populations of neighbouring connected villages, which we focus on here, is in effect just a case of such assortative mixing, but involving more circumscribed and larger populations.

Another mechanism that has been proposed to lead to stable low prevalence, included in another model of onchocerciasis spread, is that uptake of mf by flies becomes more efficient at very low prevalence [20]. If this or another unknown stabilising mechanism proves true, the break-point prevalence level for onchocerciasis can be significantly lower than we presume here, even without assuming strong heterogeneity or assortative mixing [21]. When low prevalence can be sustained within a local village population, then for elimination, implementing treatment here is clearly necessary, rather than just recommended.

Our results then underline the importance of gaining better understanding of how low prevalence settings are actually sustained. More entomological and localised knowledge on the presence of fly breeding sites, and on human traffic and/or on fly movement, could help judge what aspect is factually responsible for a given prevalence level, and so help establish the best intervention policy. For example, if assortative mixing can be identified to play an important

role, perhaps MDA can be limited or targeted to a specific high-risk subgroup of villagers. On the other hand, based on our current analyses, in case of spill-over mainly due to fly movement, MDA could be deemed fully unnecessary in many locations. With the limited current knowledge, we cannot determine a generic prevalence threshold on which to base the decision to implement MDA or not.

Demonstrating dependency on neighbouring high-transmission settings in the field would be useful to guide local elimination policy in specific situations, and to validate our work, but is challenging. Information can be obtained from long-term follow-up studies tracking changes in prevalence in neighbouring treated and untreated communities: in case of dependency, a decline in prevalence would be expected in the untreated village, as shown in Fig 4, bottom-left. Since skin snips studies on mf levels are now less common, for a comparison with such field data, future modelling work should include other infection indicators such as OV-16 antibody prevalence in children [22], or infection in flies. Additional information on general levels of connectedness can perhaps also come from large-scale genetic studies of worms and flies [9] or by studying the role of infection import in settings with post-MDA resurgence. Quantifying the degree of connectedness in specific settings will usually be difficult due to uncertainty about local transmission conditions, baseline endemicity and exact history of control.

A simplified connectivity between villages was included before in ONCHOSIM model studies, as well as in studies with the related LYMFASIM model for lymphatic filariasis; a fixed parameter function termed the external force of infection was used to explain stable low prevalence settings [23, 24]. This parameter basically corresponds to the relatively simple case of flies being blown only from a high to a low prevalence area, as one of the options modelled here. If there is MDA implemented in the high prevalence area, this inflow dries up over time, as was included by modelling a reference scenario. However, two-way contact of villages, by flies moving both ways or by humans commuting, cannot be adequately captured through an external force of infection. Therefore, the approach taken in this study can be considered a necessary step forward in understanding systems of connected villages. This understanding can also help guide when the more time-efficient model version, with the external force of infection function, sufficiently captures relevant transmission dynamics.

In conclusion, expanding treatment to lower prevalence regions may not be necessary, but could certainly speed up the road to an onchocerciasis-free world. However, at relatively low population size and very low prevalence in a location, setting up a control program may not be worth the effort, as treating locations near fly-breeding sites a few years longer could be deemed most efficient. Ideally, local knowledge and additional research should establish how low prevalences are actually sustained, in order to inform most optimal policy decisions.

## Supporting information

**S1 Text. Detailed methods with mathematical formulae. Table A in S1 Text. PRIME-NTD checklist guiding principles for policy relevant modelling. Fig A in S1 Text. Spill-over equilibrium prevalence as influenced by the type of connection between two villages. Fig B in S1 Text. Spill-over equilibrium prevalence as influenced by the local ABRs of two connected villages. Fig C in S1 Text. Spill-over prevalence as influenced by the relative size of the two villages. Fig D in S1 Text. Spill-over prevalence in case of three villages. Fig E in S1 Text. The individual variation in time spent outside the home village (as controlled by the model input parameter *M*, see the Methods section) and the resulting spill-over of prevalence. Fig F in S1 Text. Impact of MDA strategy on the probability of onchocerciasis elimination for different geographical settings. Fig G in S1 Text. Impact of MDA strategy in**

scenarios with three villages. **Fig H in S1 Text.** Impact of MDA strategy in scenarios with different MDA coverages. **Fig I in S1 Text.** Impact of MDA strategy in scenarios with different individual variation in the time spent outside the home village. **Fig J in S1 Text.** Decrease in the number of years of MDA required for 90% onchocerciasis elimination probability, by treatment implemented throughout also in the spill-over prevalence village (s) rather than just in the higher prevalence village (parameter sensitivity scenarios). **Fig K in S1 Text.** Decrease in the number of years of MDA required for 80% or 99% onchocerciasis elimination probability, by treatment implemented throughout also in the spill-over prevalence village(s) rather than just in the higher prevalence village. **Fig L in S1 Text.** Decrease in the number of years of MDA required for 90% onchocerciasis elimination probability, by treatment implemented throughout also in the spill-over prevalence village (s) rather than just in the higher prevalence village (with three village scenarios per village comparisons). **Fig M in S1 Text.** Impact of MDA duration on onchocerciasis elimination when treating inhabitants of Village 2 fewer consecutive years than those in Village 1. **Fig N in S1 Text.** Impact of MDA duration on onchocerciasis elimination when treating inhabitants of Village 2 fewer consecutive years than those in Village 1, as dependent on the timing of MDA in Village 2.
(PDF)

**S1 Model Code. R scripts to run the model and analyses.**
(ZIP)

## Author Contributions

**Conceptualization:** Anneke S. de Vos, Wilma A. Stolk, Luc E. Coffeng, Sake J. de Vlas.

**Formal analysis:** Anneke S. de Vos.

**Investigation:** Anneke S. de Vos.

**Methodology:** Anneke S. de Vos, Wilma A. Stolk, Luc E. Coffeng, Sake J. de Vlas.

**Software:** Anneke S. de Vos, Sake J. de Vlas.

**Supervision:** Luc E. Coffeng, Sake J. de Vlas.

**Writing – original draft:** Anneke S. de Vos.

**Writing – review & editing:** Wilma A. Stolk, Luc E. Coffeng, Sake J. de Vlas.

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
