## [Decision Letter · Decision Letter 0]

13 Feb 2021

Dear Dr. de Vos,

Thank you very much for submitting your manuscript "The impact of mass drug administration expansion to low onchocerciasis prevalence settings in case of connected villages" for consideration at PLOS Neglected Tropical Diseases. As with all papers reviewed by the journal, your manuscript was reviewed by members of the editorial board and by several independent reviewers. The reviewers appreciated the attention to an important topic. Based on the reviews, we are likely to accept this manuscript for publication, providing that you modify the manuscript according to the review recommendations. 

Sincerely,

Sabine Specht

Associate Editor

Sara Lustigman

Deputy Editor

Reviewer's Responses to Questions

**Key Review Criteria Required for Acceptance?**

**Methods**

-Are the objectives of the study clearly articulated with a clear testable hypothesis stated?

-Is the study design appropriate to address the stated objectives?

-Is the population clearly described and appropriate for the hypothesis being tested?

-Is the sample size sufficient to ensure adequate power to address the hypothesis being tested?

-Were correct statistical analysis used to support conclusions?

-Are there concerns about ethical or regulatory requirements being met?

Reviewer #1: The study design is appropriate, even as additional clarity is needed in some areas.

Reviewer #2: The methods are clearly explained and described. The reasoning is sound and clear. It might help, perhaps in future work, to model differential MDA coverage or lower coverage in mobile groups. It may also help to note that identifying when and how far flies are traveling is difficult and resource intensive, but much easier to do so in humans.

**Results**

-Does the analysis presented match the analysis plan?

-Are the results clearly and completely presented?

-Are the figures (Tables, Images) of sufficient quality for clarity?

Reviewer #1: The analysis is robust and the results clearly presented.

Reviewer #2: The results are generally clearly presented and framed in such a way that they will be useful for programs. Question: in lines 272-275, the text describes an average of 28% mf prevalence in lines showing elimination. This is confusing - shouldn't prevalence be much, much lower if elimination is achieved? Question: in lines 361 to 363, the text says that "..the required MDA duration when treating only Village 1 is always less than double (or triple) that of throughout treating both (or all three) villages..." Is this not backwards? Isn't the duration of treatment longer when focused only on Village 1?

**Conclusions**

-Are the conclusions supported by the data presented?

-Are the limitations of analysis clearly described?

-Do the authors discuss how these data can be helpful to advance our understanding of the topic under study?

-Is public health relevance addressed?

Reviewer #1: The authors' conclusions are supported by the data presented.

Reviewer #2: The conclusions relate directly to the data and anticipate many questions from readers. The conclusions seem to focus more on human populations, whose movement is easier to monitor and evaluate.

**Editorial and Data Presentation Modifications?**

Reviewer #1: (No Response)

Reviewer #2: For all this paper's strengths, it relies on parasitological measures and levels of prevalence that are, nowadays, not as commonly encountered in active programs. Current guidelines for interruption of transmission rely on serological and entomological endpoints, which likely correlate with mf prevalence far below that studied in this paper and those recommended by APOC. It is also known that lower levels than 20-40% mf exist and persist in absence of a traditionally hyper-endemic village nearby. Finally, programs with ongoing MDA may have suppressed their mf levels below the thresholds in this paper, but could still benefit from these results. In this reviewer's opinion, it would be helpful to address 1) how this work could be validated or supported with other indicators of transmission, and 2) present ideas for programs to utilize this approach when they either do not have baselines or have undertaken several years of MDA.

**Summary and General Comments**

Reviewer #1: This is a well-written, rigorously designed study that uses a mathematical model to explore the impact of the degree/direction of population connectedness on onchocerciasis transmission, as well as the effect of local vs. large-scale mass drug administration across a set of connected communities. The authors find that—in setting below the ABR threshold needed for stable local transmission—persistence of onchocerciasis can be explained by population movement (both human and intermediate hosts) from areas with higher ABR. They examine the theoretical constraints on elimination of onchocerciasis when MDA in undertaken in the higher prevalence, contributing area. A major strength is the formal expression and estimation of shortened MDA requirements when target populations are expanded to include the lower ABR areas. Some questions are raised over the treatment of movement in the model—the treatment of fly host movement, for instance, is not clear. What is the modeled (or implied, based on the representation in the model) time course/rate of movement? Given the short lifespan of the host, how might the effects of movement be counteracted by mortality during/following movement? Over what time period does the convergence of V1 and V2 prevalences occur (across increasing degrees of movement) in Figs 2, 3, and might the time-to-convergence be relevant to consider with respect to adaptive MDA strategies, as well as the possibility that seasonal movements may ameliorate or exacerbate the convergence? Which regions on the x-axis of figure 3 (bottom panel) are realistic/meaningful given actual host movement behavior? I understand the researchers are seeking a highly simplified model system in order to explore the effects of movement, but doing so with more robust treatment of movement dynamics may yield greater insights. Another area of concern is the potential for heterogeneity in human host movement behaviors to upend the findings of the study. It is not clear why, if movement is confined to a small number of highly mobile individuals (as is observed in many contexts), the qualitative results of the study do not change (as represented in Fig S5)? Deeper explanation and exploration is needed here, in this reviewer’s view, to make a more convincing case that heterogeneity is unimportant given prior research on connectivity and movement heterogeneity.

Reviewer #2: This paper will be important for the onchocerciasis community, as it quantifies the effects of human and fly movement on transmission. Although it would make programs easier, humans and vectors do not simply stay in one place for the duration of an elimination program. They move, and this has effects both on their home communities as well as others. MDA programs should be sensitive to these realities but often lack the tools and frameworks to understand the implications.

The traditional monitoring approach is to focus on specific (usually highest-prevalence) villages; nearby villages are either investigated as well - which can be resource intensive - or the results from the highest risk village are assumed to be true for other areas. In the worst case scenario, nearby or farther flung villages are ignored entirely, although it is clear that humans, and possibly flies, are quite comfortable moving between them. This limits the effectiveness of MDA programs. MDA can go on longer than needed, or communities are at risk of recrudescence due to previously ignored populations.

PLOS authors have the option to publish the peer review history of their article (what does this mean?). If published, this will include your full peer review and any attached files.

Reviewer #1: No

Reviewer #2: No
---

## [Decision Letter · Decision Letter 1]

26 Apr 2021

Dear Dr. de Vos,

We are pleased to inform you that your manuscript 'The impact of mass drug administration expansion to low onchocerciasis prevalence settings in case of connected villages' has been provisionally accepted for publication in PLOS Neglected Tropical Diseases.

Best regards,

Sabine Specht

Associate Editor

Sara Lustigman

Deputy Editor

Reviewer's Responses to Questions

**Key Review Criteria Required for Acceptance?**

**Methods**

-Are the objectives of the study clearly articulated with a clear testable hypothesis stated?

-Is the study design appropriate to address the stated objectives?

-Is the population clearly described and appropriate for the hypothesis being tested?

-Is the sample size sufficient to ensure adequate power to address the hypothesis being tested?

-Were correct statistical analysis used to support conclusions?

-Are there concerns about ethical or regulatory requirements being met?

Reviewer #1: The revision has substantially improved discussion of limitations of the research

Reviewer #2: The methods are clearly described and appropriate.

**Results**

-Does the analysis presented match the analysis plan?

-Are the results clearly and completely presented?

-Are the figures (Tables, Images) of sufficient quality for clarity?

Reviewer #1: Results in several areas have been revised to address some shortcomings identified by reviewers

Reviewer #2: The revisions make the results more understandable. The paper is clear and informative.

**Conclusions**

-Are the conclusions supported by the data presented?

-Are the limitations of analysis clearly described?

-Do the authors discuss how these data can be helpful to advance our understanding of the topic under study?

-Is public health relevance addressed?

Reviewer #1: (No Response)

Reviewer #2: The conclusions are appropriate and insightful.

**Editorial and Data Presentation Modifications?**

Reviewer #1: (No Response)

Reviewer #2: (No Response)

**Summary and General Comments**

Reviewer #1: The manuscript is an outstanding contribution to our understanding of movement dynamics and control of onchocerciasis through MDA

Reviewer #2: (No Response)

PLOS authors have the option to publish the peer review history of their article (what does this mean?). If published, this will include your full peer review and any attached files.

Reviewer #1: No

Reviewer #2: No

---

## [Editor Report · Acceptance letter]

6 May 2021

Dear Dr. de Vos,

We are delighted to inform you that your manuscript, "The impact of mass drug administration expansion to low onchocerciasis prevalence settings in case of connected villages," has been formally accepted for publication in PLOS Neglected Tropical Diseases.

Best regards,

Shaden Kamhawi

co-Editor-in-Chief

Paul Brindley

co-Editor-in-Chief
